# 3D Inductive Frequency Selective Structures Using Additive Manufacturing and Low-Cost Metallization

**DOI:** 10.3390/s22020552

**Published:** 2022-01-11

**Authors:** Juan Andrés Vásquez-Peralvo, Adrián Tamayo-Domínguez, Gerardo Pérez-Palomino, José Manuel Fernández-González, Thomas Wong

**Affiliations:** 1Radiation Group, Department of Signals, Systems and Radio Communications, Universidad Politécnica de Madrid, ETSI Telecomunicación, 28040 Madrid, Spain; jvasquez@gr.ssr.upm.es (J.A.V.-P.); jmfdez@gr.ssr.upm.es (J.M.F.-G.); 2Department of Electromagnetism and Circuit Theory, Universidad Politécnica de Madrid, ETSI Telecomunicación, 28040 Madrid, Spain; gerardo.perezp@upm.es; 3Illinois Institute of Technology, Chicago, IL 60616, USA; twong@ece.iit.edu

**Keywords:** frequency selective structures, additive manufacturing, metamaterials

## Abstract

The use of additive manufacturing and different metallization techniques for prototyping radio frequency components such as antennas and waveguides are rising owing to their high precision and low costs. Over time, additive manufacturing has improved so that its utilization is accepted in satellite payloads and military applications. However, there is no record of the frequency response in the millimeter-wave band for inductive 3D frequency selective structures implemented by different metallization techniques. For this reason, three different prototypes of dielectric 3D frequency selective structures working in the millimeter-wave band are designed, simulated, and manufactured using VAT photopolymerization. These prototypes are subsequently metallized using metallic paint atomization and electroplating. The manufactured prototypes have been carefully selected, considering their design complexity, starting with the simplest, the square aperture, the medium complexity, the woodpile structure, and the most complex, the torus structure. Then, each structure is measured before and after the metallization process using a measurement bench. The metallization used for the measurement is nickel spray flowed by the copper electroplating. For the electroplating, a detailed table showing the total area to be metallized and the current applied is also provided. Finally, the effectiveness of both metallization techniques is compared with the simulations performed using CST Microwave Studio. Results indicate that a shifted and reduced band-pass is obtained in some structures. On the other hand, for very complex structures, as in the torus case, band-pass with lower loss is obtained using copper electroplating, thus allowing the manufacturing of inductive 3D frequency selective structures in the millimeter-wave band at a low cost.

## 1. Introduction

Frequency selective surfaces are electromagnetic filters whose frequency response depends on the incidence angle and polarization of the impinging electromagnetic waves [1]. These filters include periodic elements that will act as a low-pass, high-pass, band-stop, or band-pass filter depending on their arrangement on the structure. Munk has extensively described the operation principle of these structures in his book [2]. In his work, the author suggests that, when an electromagnetic wave impinges over an FSS, it excites currents on the periodic elements. The amplitude of the induced currents depends mainly on the coupling produced by the electromagnetic wave and the geometry of the periodic elements. As a result, the induced currents become an electromagnetic source itself, creating a scattered field that generates the entire field of the structure, together with the incident electromagnetic wave. Consequently, modifying unit cell parameters like substrate properties, resonant element arrangement, or extending the conventional 2D structure to a 3D structure will modify its frequency response.

3D FSS are generally two-dimensional periodic arrays whose resonant elements, placed in the *x*-*y* plane, extend along the *z*-axis. This extension modifies the current distribution in the structure, which changes its frequency response to such an extent that specific 2D arrays that work as a band-pass filter become a band-stop filter when transformed into a 3D array [3]. The use of 3D FSS structures has certain advantages and disadvantages over the 2D FSS structures. The most notable benefits are more degrees of freedom in spatial filter design, excellent mechanical stability, strength, and, in some cases, better angular stability at different angles of incidence [4]. On the other hand, these structures present some drawbacks, such as complex manufacturing, high production costs, and unsuitable for limited space applications.

3D FSS manufacturing can be challenging, especially in choosing an appropriate fabrication technique. Literature shows that 3D FSS prototypes are mainly fabricated using three different techniques.

The first is by mechanical manufacturing through milling or metal bending. One of the most relevant works using this technique is presented in [3]. The authors propose a 3D FSS that works in L, S, and C bands, consisting of hollow metallic cylinders. These cylinders are manufactured by mechanical processes and placed on a foam surface for measurement. The same author also proposes a similar prototype with springs [5], while others propose hollow cylinders with rollers [6]; both works explore reconfigurability. As described above, the array elements are fabricated individually and placed manually on a dielectric surface. This arrangement makes the array expensive, error-prone, and not very mechanically stable.

The second manufacturing technique is a three-dimensional array of 2D FSS fabricated by etching, milling, or laser techniques. This type of manufacturing is presented in [7], where the authors design and measure a 3D FSS that works as a band-stop filter. The 3D model proposed by the authors consists of short periodic microstrip lines etched over an RT Duroid 6010 substrate, vertically opposed and also periodically arranged. The same authors propose a 3D FSS that works as a band-pass filter using the same manufacturing technique [8]. Since this type of FSS offers high miniaturization, its angular stability at different incidence angles is high. On the other hand, manufacturing this FSS has certain disadvantages, such as high manufacturing costs and fabrication errors, as each strip containing the microstrip lines has to be manufactured separately and manually assembled at the end.

The last, but no less important, manufacturing technique is additive manufacturing. In additive manufacturing of FSS, the metallization of the conductive area has been done by different methods. The works presented in [9,10,11,12,13,14] use the additive manufacturing method called material extrusion. In this method, thermoplastic materials such as Polylactic Acid (PLA) or Acrylonitrile Butadiene Styrene (ABS) are melted and extruded through a nozzle, forming the desired geometric figure. To provide the necessary metallization, the authors have chosen to manually paint the structure with conductive paint, usually silver-based, by brush or spray. Other researchers use two 3D printers that work with material extrusion, one to make the substrate and the second to cover some areas with a metal surface. The second printer uses silver-based compounds [15] or carbon-fiber-reinforced thermoplastic [16] to inject the metal layers needed for the final assembly onto the previously printed dielectric surface. Because the material extrusion method generates a prototype with high roughness, the VAT Photopolymerization, specifically the stereolithography (SLA), is a trending method due to its new improvement in stitching quality [17], curing process [18], energy consumption [19], and graphene-oxide use capabilities [20]. With VAT photopolymerization, some authors manufacture the dielectric and use vacuum deposition technology for metallization [21]. In [22], the authors use the electric discharge machining technique to manufacture a polarizer based on all-metal 3D FSS structures. Finally, in [23], the authors design conformal structures, where certain sections must be metallized. They use the Water Transfer Printing (WTP) technology, which consists of printing the FSS structure on polyvinyl alcohol (PVA) with any conductive ink. This film is then placed on water until the PVA is completely dissolved, and only the structure to be transferred floats. The final step is to slowly immerse a 3D mold in the water until the metal layer adheres to it completely.

This work shows the feasibility of building low-cost millimeter-wave band inductive 3D frequency selective structures (I3DFSSs) using additive manufacturing and two different metallization techniques. For this purpose, three different types of 3DFSS have been designed and manufactured using additive manufacturing.

This paper is organized as follows: Section 2 introduces the design and simulation of the three frequency selective structures. In addition, two metallization processes, nickel spray and copper electroplating, are described, including its application to the square aperture, woodpile, and torus structure. The measurement process and the comparison of the results for the three prototypes are presented in Section 3. Section 4 presents a discussion of the results of the submitted paper. Section 5 presents the conclusions obtained and future work proposed by the authors.

## 2. Design and Manufacturing

Three 3D FSS, shown in Figure 1, have been developed for use in the V-band. These structures differ in their design and manufacturing complexity so that the effectiveness of nickel spray and copper electroplating can be tested. The first structure is the square 3D aperture. This structure is the easiest to manufacture as it is created by extruding the 2D design in the *z*-axis. The second structure is the woodpile, which is more complex than the square aperture. This structure consists of filaments stacked on top of each other, which creates an excellent mechanical balance. Finally, the third structure is the torus, whose complexity is higher than the previous two. The torus structure consists of 3D rings, rotated 90 degrees and self-intertwined when extruded across the *z*-axis. Subsequently, the frequency response of the structures is then analyzed as a function of their extension through the *z*-axis.

The unit cells are simulated using the frequency domain solver with Floquet ports available in CST Microwave Studio. The mesh selected uses tetrahedrons to have a conformed mesh with the 3D structure. Periodicity conditions have been applied to replicate the structure on the *x* and *y*-axis.

### 2.1. Square Aperture 3D FSS

The resonant frequency of the 2D square aperture is very close to the onset of grating lobes, which makes it one of the least preferred structures in the design of filters, except in filter synthesis or artificial magnetic conductors. By extending the structure on the *z*-axis, which we call *t*, the band-pass shifts down in frequency from the grating lobe region, as illustrated in Figure 2. An additional effect is a high roll-off near 50 GHz, which increases with thickness. The previous results can be easily explained by analyzing the unit cell as a waveguide. Under this consideration, the waveguide itself is considered as a high-pass filter, determined by its cut-off frequency in TE-mode using (Equation 1):(1)$$f_{C_{10}}=\frac{c}{2a}$$
where *c* is the speed of light, and *a* is the width of the grating trough. The result of the previous formula is the same frequency at which the band-pass of the 3D structure occurs.

An additional effect of increasing *t* is the onset of new resonances. Figure 3 illustrates the reflection coefficient as a function of *t* and frequency. The results show that, as *t* increases in multiples of $\lambda_{0}/2$, new decreasing resonances, represented by bluish lines, appear and settle to an asymptotic value. Furthermore, each resonance has its own bandwidth, represented in Figure 3 by −10 dB notation. When *t* is increased, these resonances are arranged so that their −10 dB bandwidth is blended and form an enhanced band-pass. The previous can be verified in Figure 2 and Figure 3 at t=10 mm and in the frequency range from 53.6 to 61.6 GHz. These new resonances and the improved band-pass effect are easily explained using the equivalent circuit of a square aperture presented in Figure 3. As *t* increases, new replicas of the original shunt LC circuit appear, generating new resonances; meanwhile, the new transmission lines with βl act as impedance inverters. Thus, by controlling *t*, we can obtain a structure that provides a reflection coefficient with high flatness or equal ripple response.

#### 2.1.1. WoodPile FSS

The woodpile structure has been mainly used as a dielectric EBG and presented in [24,25]. In these works, the authors design and manufacture fully dielectric woodpile structures operating in the G and X bands, respectively. In the present work, a metallic woodpile structure has been simulated with up to six stacking stages to ensure complete symmetry in all directions. In addition, a frequency analysis was performed considering the six stages is obtained to understand the effect of stacking layers, as illustrated in Figure 4. As a result of the stacking stages, the selectivity increases and reflects any electromagnetic wave at normal incidence up to 50 GHz.

One problem with using this structure as an IFSS is the closeness of the resonant frequency to the grating lobe region, which leads to poor angular stability, as will be seen later in this publication.

#### 2.1.2. Torus

The torus structure is the most complex compared to the previous two. At each design stage, a torus ring intersects with a copy of itself rotated $90^{\circ }$ at each of its cardinal points. The new rings are in turn intersected by new rings which, like the previous ones, are rotated by $90^{\circ }$. This process is repeated depending on the number of phases. In this case, the torus structure is designed using four stages. The simulations illustrated in Figure 5 show that, as the number of stages increases, the roll-off is more pronounced, as in the 3D square aperture and the woodpile structure, which allows transmission from 43 to 53 GHz. Another interesting result shows that the band-pass of this structure is lowest below the grating lobe region. This phenomenon is not new because the interwoven structures configured in 2D or 3D allow strong miniaturization due to the increase in the equivalent capacitance and inductance.

### 2.2. Manufacturing

The manufacturing of the three prototypes, described and analyzed in the previous section was carried out in four different phases.

#### Full Structure Design

In this phase, the complete structures are designed using the unit cells obtained in the previous section. The height and width dimensions of the prototypes are set to 140 mm × 140 mm, approximately 5% smaller than the maximum allowed by the Form 3 3D printer to leave room for the structure base. The square aperture structure has 30 × 30 unit cells, the woodpile structure has 25 × 25 unit cells, and the torus structure has 35 × 35 unit cells. The dimensions of the designs meet the criterion of more than 20 unit cells in length and in width to guarantee a frequency response similar to that of a structure extending to infinity on the *x* and *y*-axis [26]. To avoid erroneous measurements, the edges of the designs are frameless.

### 2.3. Additive Manufacturing

The complete structures are manufactured using a Form 3 by Form Labs, which uses SLA technology and has a resolution of 25 um on the *x*, *y*, and *z*-axis, which is more than sufficient for a very accurate replica of the design. The structures were fabricated using Gray resin, characterized in a waveguide WR-28, and showed a dielectric constant $\epsilon_{r}$ = 2.738 and tanδ = 0.02 @ 40 GHz. The woodpile and torus structures were printed vertically for mechanical stability during printing, while the square aperture structure was printed horizontally. In addition to the prototype, the 3D printer adds support posts to hold some design elements during printing. Once printing is complete, the structures are immersed in isopropyl alcohol (IPA) for approximately 30 min. The bath in IPA removes the excess resin that did not cure during printing and removes any possible blockages near the details of the structure. Finally, the design is cured with UV rays at $35^{\circ }$ for an hour, giving the design greater strength. The temperature could be higher and the curing time shorter, but this will deform the flatness of the structures. The final step is to remove the holding posts. Depending on the design, removing these supports is more time-consuming and delicate, with the torus structure being the most complicated and the square aperture the easiest. The finished 3D dielectric FSSs are illustrated in Figure 6a–c.

### 2.4. Nickel Spray Painting Metallization

Both metallization processes described in this article use the process described in [27]. Below is a brief description of the steps for both methods.

Nickel spray metallization has two purposes. The first is to prepare the structure for measurements corresponding to metallization type 1; the second is to create a layer of conductive material on the surface of the structure for metallization by copper electroplating. The designs are metallized using super shield spray, manufactured by MG Chemicals. This spray contains nickel particles that adhere to the surface of the structure and form a film of approximately 100–150 μm. The curing process of this paint takes about 24 h at room temperature. Figure 6d–f show the prototypes that were metallized with the spray and cured for 24 h.

### 2.5. Copper Electroplating

One advantage of using inductive FSS (IFSS) is that the entire structure is electrically conductive at every point, which facilitates its metallization by electroplating. To do this, we immerse our 3DIFSSs, called cathodes in this process, in an electrically conductive solution, in our case based on copper sulfate, sulfuric acid, and some extra additives to smooth the copper deposition. For more details on this process, see [27]. One or more copper plates, called anodes in this process, are also immersed in this solution without touching the cathode. Both anode and cathode are connected to a current source. When a current is applied, the ions in the copper sulfate solution move towards the cathode; this process is called reduction. At the same time, the anode loses electrons and dissolves copper ions in the solution; this process is called oxidation. When this process is repeated, copper ions are deposited on the cathode and metallize all conductive areas. The metallization times depend on the applied current, the immersed cathode area, and the concentration of the copper sulfate solution. Figure 6g–i show the three metallized structures using copper electroplating according to the times and currents given in Table 1.

## 3. Measurements

### 3.1. Measurement Setup

The manufactured structures are characterized using a measurement bench, consisting of two WR-20 horns with their respective transitions, a support with a frame to hold the DUT, and an analyzer ANRITSU 3739C. A schematic representation of the measurement set-up is shown in Figure 7a. The antennas are placed at a distance of 120 mm, from the surface of the structure to ensure an illumination taper of −18 dB, sufficient to correctly characterize the structure. The DUT support has a fixed rotation but allows for changing the height of the DUT over the bench. The analyzer is configured with a resolution of 512 points in the frequency range of 50 to 75 GHz and uses a local oscillator of 100 Hz. The instrument was calibrated using the transmission, reflection, and match technique (TRM). The reflection is performed using a metal plate located on the DUT support and the transmission without a metal plate. The actual test bench is illustrated in Figure 7b.

The manufactured and metallized structures were characterized using the S-parameters determined at the measurement bench. Each 3DFSS was measured in three stages: unmetallized (dielectric), nickel sprayed, and copper electroplating. Since this work aims to verify both metallization methods at millimeter frequencies, the results were obtained only at normal incidence and, due to the symmetry of the three structures, only the results of the TE-mode are presented.

### 3.2. 3D Dielectric FSS

Figure 8 illustrates the results of the S-parameters measured in each of the printed dielectric structures. In the case of the dielectric square aperture, a very narrow band-gap appears in the simulation results at a center frequency of 57.9 GHz, while the measurement showed a center frequency of 58.12 GHz. The woodpile structure presents a band-gap at a 53.6 and 54.2 GHz center frequency for simulation and measurements, respectively. Last but not least, the simulated torus structure gives a band-gap at a center frequency of 58.94 GHz and its measurement at 57.65 GHz. For the previous structure, the onset of grating lobes in the measurement has been shifted by approximately 3.7 GHz, compared to the simulations. Therefore, only in this case, the band-gap is not as pronounced as in the previous structures.

As can be seen from the results obtained previously, there are some discrepancies between the measured and simulated results, which can be attributed to several reasons. The first and most obvious reason are the imperfections caused by the support posts printed in the manufacturing process and visible in Figure 7. These imperfections affect the frequency response, especially for complex structures such as the woodpile and the torus. Another effect that causes such discrepancies is the buckling of the structures. In addition, it is known that one of the problems with STL manufacturing is that the dielectric tends to deform during the curing and drying process, especially with thin thicknesses and planar designs as in this case. Furthermore, the band-gap of dielectric FSS is known to be very sensitive to incidence angles, which explains the discrepancies especially in Figure 8b,c. Finally, another factor affecting the measurements is the high losses of the dielectric, which are significantly higher at these frequencies than in its original characterization.

### 3.3. 3D Metallized FSS

For a fair comparison between the measured and simulation results, some adjustments have been made in the last one. First, the models are modified, adding an extra thickness of 150 μm and 170 μm for the nickel spray and copper electroplating metallization, respectively. This thickness was estimated considering the structure weights before and after the metallization phases and the original effective area. Second, the materials are adjusted using nickel and copper with proper rugosity for each metallization process. The measurements for the square aperture, woodpile, and torus structures using nickel spray and cooper electroplating are compared with the simulated and are illustrated in Figure 9.

## 4. Discussion

Figure 9a shows the results of the reflection coefficient measured on the square aperture structure. The results show that both metallizations generate a similar band-pass compared to the simulation. On closer inspection, the bandpass is slightly lower for the nickel spray metallization than for the copper electroplating, to the point where the reflections at 58 GHz are above −10 dB. On the other hand, the copper electroplating shows a continuous band-pass of 4.85 GHz, 23.13% lower than in the simulation. This bandwidth reduction for both metallization techniques occurs due to the prototype curvature, which generates a bandwidth reduction due to the onset of the first grating lobe placed around 65 GHz. As expected, the transmission coefficient results, illustrated in Figure 9, show that the losses obtained using copper electroplating are lower than the nickel spray. In addition, the band-pass has suffered a slight frequency shift, attributed to minor measurement errors, post residuals, and deformation of the structure during the curing process.

For the woodpile structure, the reflection coefficient is illustrated in Figure 9b. The results show that the band-pass is minimal, shifted up in frequency for both measured prototypes. The nickel spray fails to generate a band-pass below −10 dB, while the copper electroplated achieves a clear band-pass around 58.5 GHz. The previous is attributed to the higher conductivity of the copper compared to nickel and the metallization uniformity. The frequency shift in both structures is mainly due to the previously analyzed curvature and the rounding in the corners due to paint accumulation. Transmission results, illustrated in Figure 9e, show higher losses in the case of nickel spray metallization than in copper electroplating, as expected. Besides the low conductivity of nickel, these high losses and reduced band-pass are attributed to the increased angle of incidence sensitivity due to the structure’s early onset of grating lobes. Finally, Figure 9c shows the results of the reflection coefficient of the torus structure. In this case, the simulated and measured results are in good agreement, especially for the copper electroplating. This similarity is explained by the fact that the torus structure, being an interwoven structure, provides better angular stability than in previous cases, making it more robust to incidence angles’ variations. Additionally, the resonance generated at approximately 52 GHz for the nickel spray metallization has shifted to 54 GHz for the electroplating case. This shift is due to the additional layer that metallizes the structure using electroplating. Although the generated layer is relatively thin, it has a significant effect due to the interwoven arrangement of the design. The transmission coefficient results, shown in Figure 9, show the same trend as the previous measurement. That is, the losses for nickel spray metallization are higher than for copper electroplating.

## 5. Conclusions

The study has shown the feasibility of manufacturing 3D inductive frequency selective structures at a low cost. These structures have been manufactured using additive techniques and, subsequently, metallized using nickel spray and copper electroplating. The prototypes have been characterized using a measurement bench in each manufacturing and metallization phase. The measurement results and the simulations agree reasonably well. It has also been demonstrated that the prototypes present lower losses when metallized using copper electroplating due to copper’s higher conductivity than nickel. Therefore, the extra step of copper electroplating is highly recommended for future prototypes. Furthermore, we have found a significant disadvantage of SLA printing: the buckling of the prototypes due to the curing process. This deformation changes the frequency response, especially in the working frequency band of this research. Therefore, different curing methods or resins that do not need heat for curing the material should be considered for future prototypes. Finally, this manufacturing and metallization process allows the manufacture of 3D frequency selective structures such as two-layer dichroic sub-reflectors, polarizers, and radomes at lower costs than other manufacturing and metallization methods.

## Figures and Tables

**Figure 1 sensors-22-00552-f001:**
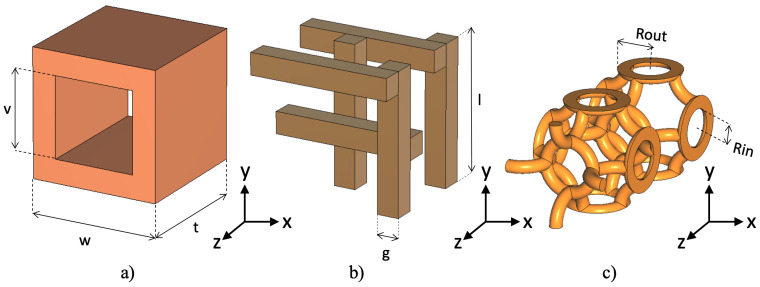
3D FSS unit cells designed to work in the V-band. (**a**) 3D square aperture FSS, using the following dimensions: w=4.5 mm, t=4.7 mm, v= 3 mm; (**b**) intertwined Woodpile FSS, using the following dimensions: g=0.85 mm, l=5 mm; (**c**) intertwined Thorus FSS, using the following dimensions: Rin=0.9 mm, Rout=1.5 mm.

**Figure 2 sensors-22-00552-f002:**
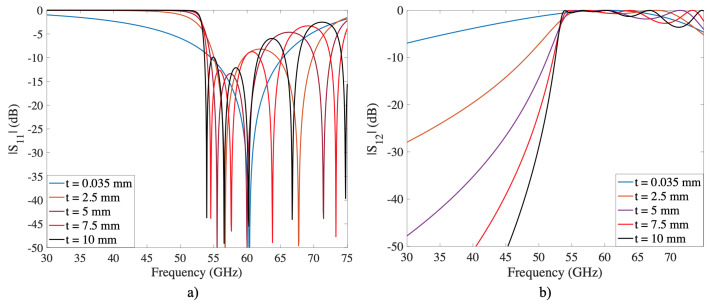
S-Parameters of an inductive grid at normal incidence and TE-mode. The thickness of the structure is increased from 0.035 to 10 mm. (**a**) reflection coefficient; (**b**) transmission coefficient.

**Figure 3 sensors-22-00552-f003:**
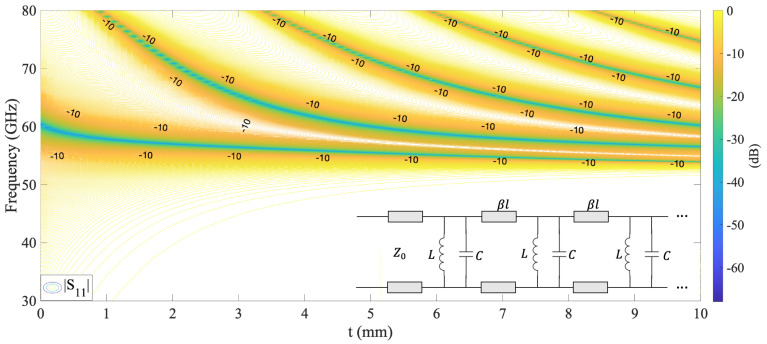
Reflection coefficient as a function of *t* and frequency, and the equivalent circuit of the square aperture illustrated in Figure 1a. The new resonances represented by shunt LC circuits appear as *t* increases.

**Figure 4 sensors-22-00552-f004:**
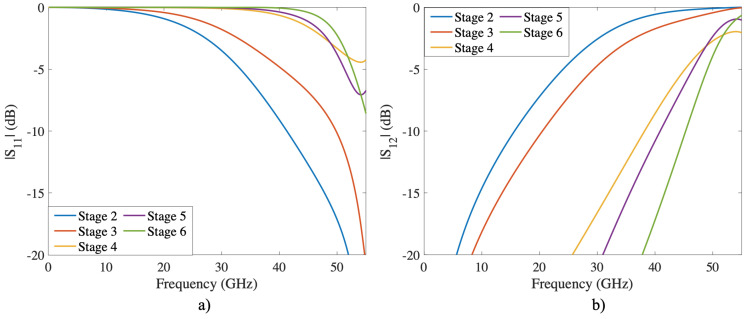
S-Parameters of the Woodpile structure at normal incidence and TE-mode. The staking stage is increased from two to six through the *z*-axis. (**a**) reflection coefficient; (**b**) transmission coefficient.

**Figure 5 sensors-22-00552-f005:**
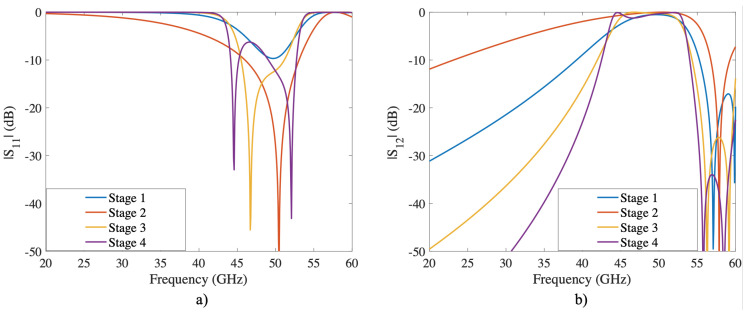
S-Parameters of the Woodpile structure at normal incidence and TE-mode. The intertwined stage is increased from one to four through the *z*-axis. (**a**) reflection coefficient; (**b**) transmission coefficient.

**Figure 6 sensors-22-00552-f006:**
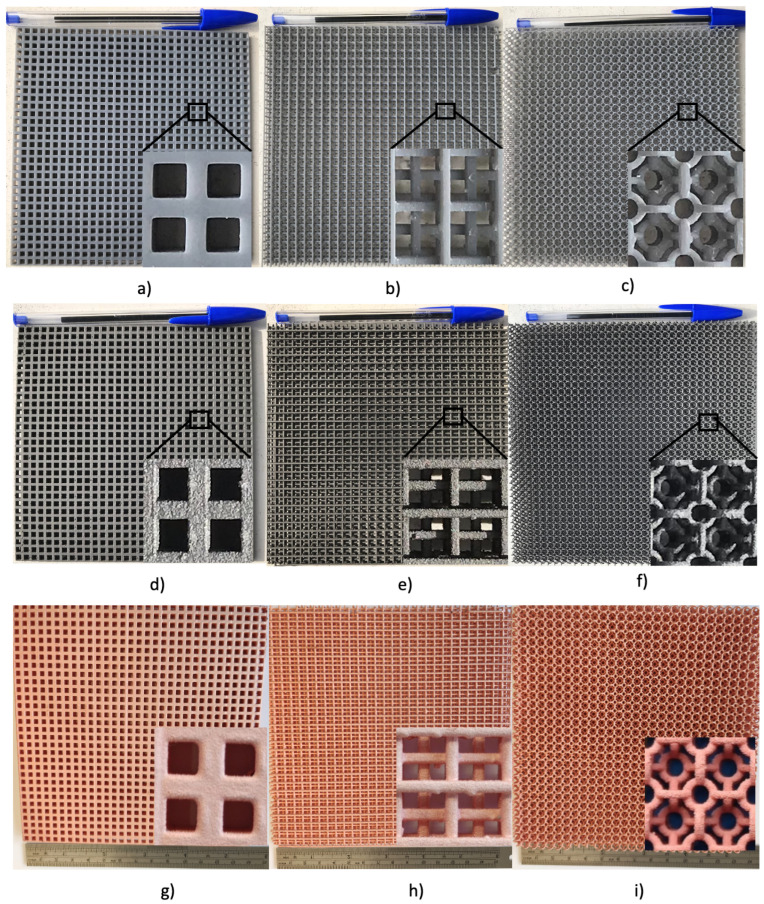
3D IFSS prototypes in each of their manufacturing stages. (**a**) square grid dielectric 3DFSS; (**b**) woodpile grid dielectric 3DFSS; (**c**) torus grid dielectric 3DFSS; (**d**) square grid 3DIFSS with nickel spray; (**e**) woodpile grid 3DIFSS with nickel spray; (**f**) torus grid 3DIFSS with nickel spray; (**g**) copper-electroplated square grid 3DIFSS; (**h**) copper-electroplated woodpile grid 3DIFSS; (**i**) copper-electroplated torus grid 3DIFSS.

**Figure 7 sensors-22-00552-f007:**
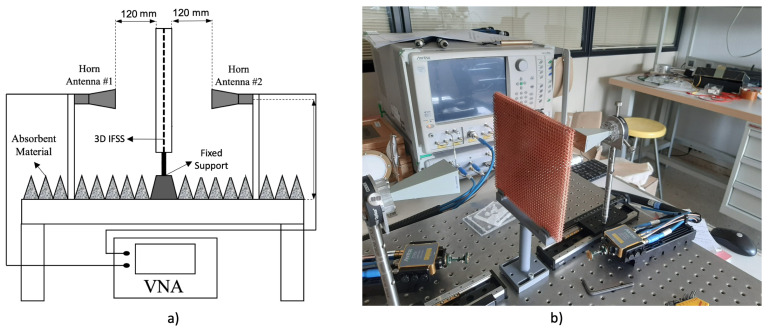
Measurement setup used to characterize the 3D FSS structures. (**a**) schematic representation of the measurement setup including distances and the names of the elements; (**b**) actual photo of the measurement setup previous to the measurement of the torus 3D IFSS.

**Figure 8 sensors-22-00552-f008:**
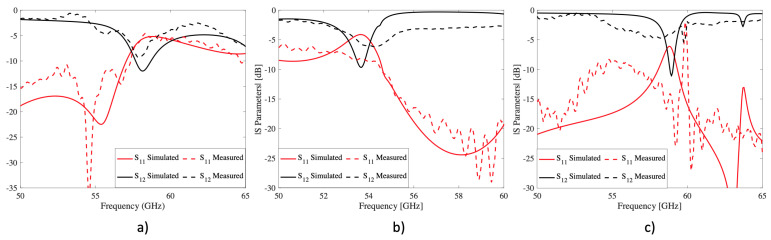
Simulated and measured transmission and reflection coefficients corresponding to the 3D dielectric FSS. (**a**) square aperture; (**b**) six-layer woodpile; (**c**) four-layer torus.

**Figure 9 sensors-22-00552-f009:**
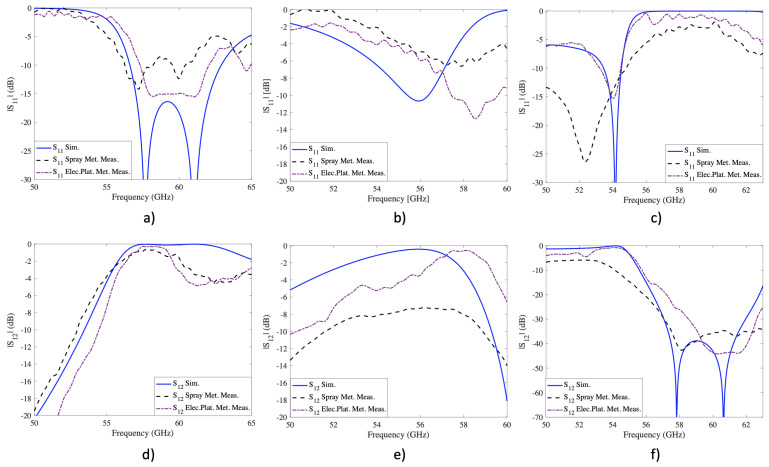
Simulated and measured reflection and transmission coefficients of metallized structures using nickel spray and copper electroplating. (**a**) reflection coefficient of the square aperture structure; (**b**) reflection coefficient of the woodpile structure; (**c**) reflection coefficient of the torus structure; (**d**) transmission coefficient of the square aperture structure; (**e**) transmission coefficient of the Woodpile structure; (**f**) transmission coefficient of the torus structure.

**Table 1 sensors-22-00552-t001:** Current and time values for copper electroplating the square aperture, woodpile, and torus structures.

	Square Aperture	Woodpile	Torus
**Area (cm${}^{2}$)**	760.27	885.35	1002.69
**Time (h)**	4	4	4
**Current (A)**	3.8	4.42	5

## Data Availability

All the data can be found within this paper.

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
