# Peer review of "3D Inductive Frequency Selective Structures Using Additive Manufacturing and Low-Cost Metallization"

_sensors, 2022, doi:10.3390/s22020552_

Round 1

Reviewer 1 Report

The first major concern is about the typos, lack of consistency of variables, incorrect section and sub-section numbering, unmatching parametric range, etc.  Also, English language and grammar need to be thoroughly revised. Secondly, the presentation of technical information is incomplete, confusing, and sometimes erroneous. There are huge mismatches between measured and simulated results which prove that the proposed manufacturing method is not suitable for mm-wave frequency applications, and for complex geometries.

Some of the corrections are indicated below.

 (1) Line 6: "......we manufacture......." in fact, it appears that the authors directly manufactured some random structures withhout optimizing them in simulation.

(2) Line 15: "band-passs". Extra 's' to be removed. Still it should be corrected as "band-pass response" throughout the paper

(3) Line 31: "electrical properties" - mention which all.

(4) Line 35: “Z-axis” This is relative and applicable only if the substrate is lying in the XY-plane.

(5) Line 36: “some” is not appropriate here

(6) Line 38: “2D FSS” To add ‘structure’

(7) Line 39: “spatial filters design”. To correct as ‘spatial filter design’

(8) Line 44: “these structures”. Replace ‘these’ with ‘3D FSS’

(9) Line : 49: “constituting” to be replaced with “comprising”

(10) Line 70: “PLA or ABS” . To be expanded.

 (11) Line 73: “other authors”. To be replaced with “other researchers”

(12) Line 94: “In addition,……” Correct the sentence.

(13) Line 97: Remove “Finally”

(14) Line 100: The heading “Materials and methods” is to be replaced with something more appropriate.

(15) Line 103: “…metallization techniques”. To be mentioned which all.

(16) Line 105 and 119: Replace “through” with “in”.

(17) Line 111 and 115: X, Y and Z –axes must be shown in Fig.1

(18) Line 120 : Variable th is not a suitable one. Use standard ones – t, h or d.

(19) Line 120: “grating lobe region as illustrated in Fig.2”. Not clear what the authors mean.

(20) Eq.(1) what is a with respect to Fig.1 ?

(21) Fig.2: Show maximum frequency upto 75 GHz

(22) Fig.2 caption: 0.035 to 10 mm is mentioned. But in Fig., 2.526 to 10 mm is shown.

(23) Line 129: Use proper symbol for lambda0

(23) Line 130: “Equivalent circuit” is not referenced to Fig.3.

(24) Fig.3: In the caption, equivalent circuit must be mentioned as inset.

(25) Fig.3 : Not clear what these individual curves represent ? What are these -10 on every curve ?. Also why it is need to show those color grading ? Need to explain in the text how to read this graph.

(26) Line 133: “Therefore controlling”. To be corrected as “Therefore by controlling”

(27) Fig.4: Show X-axis upto 75 GHz.

(28) Line 145: Not clear what the grating lobe is .

(29) Add figures like Fig.3 for WoodPile and Torus structures also.

(30) Line 218: Explain how the authors managed to use an optical bench for microwave measurement!

(31) Line 221: Antenna are 150 mm apart. But in Fig.7, they are 120 mm apart ?

(32) Line 222 :”tapper” to be corrected as “taper”

 (33) Fig.8 caption : (b) repeats

 (34) Line 281: “…results show that tThe…..” Remove ‘t’ before The

(35) Line 307:  “The results were compared favorably with the simulations”. In Fig.8 and 9 there are huge mismatches between simulated and measured results showing this manufacturing method is not useful for mm-wave frequencies. This contradicts the whole point of the paper.

Reviewer 2 Report

The manuscript entitled “3D Inductive Frequency Selective Structures Using Additive

Manufacturing and Low Cost Metalization” dealing with VPP was refereed. The author should consider my comments before I arrive at the final decision.

  1. Expand the abstract and add some more materials regarding the results.
  2. Add dimensions to Figure 1.
  3. Proofread the paper and polish the typos.
  4. The explanation on Figure 2 needs to be expanded. In the current format this is too short.
  5. Read and add the following reference to update the introduction with the new papers in SLA.
  • Uysal, E., M. Çakir, and B. Ekici, Graphene oxide/epoxy acrylate nanocomposite production via SLA and importance of graphene oxide surface modification for mechanical properties
  • An investigation on energy consumption and part quality of stereolithography apparatus manufactured parts
  • Evaluation of UV post-curing depth for homogenous cross-linking of stereolithography parts
  • Tilting separation simulation and theory verification of mask projection stereolithography process
  • Design and optimization of projection stereolithography additive manufacturing system with multi-pass scanning

  1. The paper is robust with adding future directions.

Round 2

Reviewer 1 Report

My comments are more or less satisfactorily addressed by the authors. So the paper may be accepted for publication only after a final English check.

Reviewer 2 Report

The paper is ready to go.
